# Method to Develop Legs for Underwater Robots: From Multibody Dynamics with Experimental Data to Mechatronic Implementation

**DOI:** 10.3390/s22218462

**Published:** 2022-11-03

**Authors:** Miguel Ángel Pérez Bayas, Juan Cely, Avishai Sintov, Cecilia E. García Cena, Roque Saltaren

**Affiliations:** 1Centre for Automation and Robotics UPM-CSIC, Universidad Politécnica de Madrid, 28006 Madrid, Spain; 2Escuela Superior Politécnica de Chimborazo, Riobamba 060155, Ecuador; 3School of Mechanical Engineering, Tel Aviv University, Tel Aviv 6997801, Israel; 4Centre for Automation and Robotics UPM-CSIC, Escuela Técnica Superior de Ingeniería y Diseño Industrial ETSIDI-Universidad Politécnica de Madrid, Ronda de Valencia 3, 28012 Madrid, Spain

**Keywords:** gait, quadrupedal, underwater, stiffness, damping, friction, multibody, testbed

## Abstract

Exploration of the seabed may be complex, and different parameters must be considered for a robotic system to achieve tasks in this environment, such as soil characteristics, seabed gait, and hydrodynamic force in this extreme environment. This paper presents a gait simulation of a quadrupedal robot used on a typical terrigenous sediment seabed, considering the mechanical properties of the type of soil, stiffness, and damping and friction coefficients, referenced with the specialized literature and applied in a computational multibody model with many experimental data in a specific underwater environment to avoi hydrodynamic effects. The requirements of the positions and torque in the robot’s active joints are presented in accordance with a 5R mechanism for the leg and the natural pattern shown in the gait of a dog on the ground. These simulation results are helpful for the design of a testbed, with a leg prototype and its respective hardware and software architecture and a subsequent comparison with the real results.

## 1. Introduction

Many devices have been modeled and manufactured for underwater exploration based on two main concepts: autonomy and remote operation. Autonomy is achieved with autonomous underwater vehicles (AUVs) [1] for survey tasks, seabed mapping, identification/inspection, rescue, etc., and remote operation with remotely operated vehicles (ROVs) [2] for observation, marine sampling, photography, etc. At present, other concepts have been deisnged for this task, such as legged devices, which offer other capabilities to achieve underwater exploration, including the two mentioned concepts and the idea of bio-inspired mechanisms, which are usually designed in accordance with underwater gait stability using hexapod anatomy.

In [3], a hexapod robot walking underwater considering hydrodynamics effects to improve the stability margin was considered. Picardi [4] reported a bioinspired hexapod robot with two locomotion modes: hopping and walking, where the leg motors reached the position of the next state, depending on which a momentum around a vertical axis could be generated, resulting in a rotation movement. A simulation of the mechanical modeling, seafloor environment setting, and gait planning for a hexapod robot was presented by Liu [5], who verified the correctness of movement using a virtual model. Wang Z. [6] reported a model and simulation of an underwater hexapod take in account the kinematic, dynamic, and hydrodynamics effects using an isolated body method for a virtual model. The principal parameters than influence the underwater gait were presented by Wang G. [7], with a method to compute join torque, motion parameters, hydrodynamic force, and modeling of foot-terrain force.

Many types of methods suitable for many tasks have been developed concerning quadrupedal robots. Katz [8] presented a system with back driveable modular actuators, which enabld high-bandwidth force control. Hutter [9] reported joint modules with integrated electronics that precisely control the torque. In both cases, the controllability, robustness, and importance of the motor–gearbox group during locomotion and the control of the angular position of the active joints were demonstrated, in addition to showing the kinematic, dynamic, and control strategies used. These terrestrial developments offers a different perspective for underwater exploration, with the respective technologies required, and performing (or improving) the same actions as AUVs and ROVs in inspection, sampling, and manipulation tasks in seafloor environments.

In this paper, a novel implementation based on quadruped anatomy is proposed. It has updates to its hardware, software, and movement control for terrestrial environment and offers an initial product for the underwater gait of a quadrupedal system. The quadruped robot can achieve different movement states (standby, walking, trotting, jumping, etc.) when performing its tasks, but in this study, only the normal gait state was analyzed.

Accordingly, this paper presents a simulation of the gait of a quadruped robotic system on seabed soil that must meet the required criteria regarding stiffness, contact damping, and static and dynamic friction in a specific environment with a terrigenous sediment seabed with its mechanical properties. We delimited the environment at a given depth of up to 10 m to avoid the effects of hydrostatic pressure [10], and incorporated the robotic mechanism in the soil model into the multibody simulation. We also included a previous analysis of the dynamic characteristics of the seabed with a simple mass–damping–spring–friction system, and then used these results to determine the torque required to overcome these seabed features and the quadrupedal gait state. The underwater forces’ effects were not analyzed in this study, as the gait was achieved in a linear velocity lower than 0.2 m/s to mitigate the hydrodynamic damping effects (linear and quadratic), as mentioned by Fossen [11] for dynamic positioning speed regimes <2 m/s.

Often, a computational model depends on experimental data to complement and validate it. In the case in this study, through a method that shows all the phases for its implementation, from CAD design, kinematics, dynamic simulation of its motion and environment, and the implementation of control algorithms, we created a more realistic and easier-to-validate device that uses the appropriate hardware and software, with the model of its motor–gearbox group based on the simulation results for its prototyping and implementation with the $Q_{1}$ and $Q_{2}$ angular positions measured on the testbed.

The remainder of this paper is structured as follows: Section 2 presents the robot parameters, mechanical characteristics of the quadrupedal testbed robot, mechanism in each leg, geometry, and principal dimensions, as well as the inverse kinematics of the leg mechanism. Section 3 presents the seabed model, with the stiffness, damping, and friction coefficients calculated and represented as a block diagram in Simulink^®^/SimscapeTM and another dynamic model created in Simulink^®^ to calculate the deep displacement in the same environment. Section 4 presents the prototype design with the gait simulation and results for the torque and angular position for each active joint in the simulation, gearbox design according to these results, and a leg prototype for a testbed, with the hardware and software architecture used for the low-level control of the motors. Section 5 presents the angular position and velocity (in PWM values) results for each active joint for the leg prototype in a testbed experimental gait, and the simulation results are compared with their equivalent mean squared error (MSE). Section 6 presents a discussion of the results. Section 7 presents the conclusions and recommendations for future studies.

All results will be used in future studies to develop a complete robotic system where other control algorithms can be tested and more easily validated in a real underwater test environment.

## 2. Robot Parameters

### 2.1. Mechanical Characteristics

The mechanical design was created in Autodesk Inventor^®^ and transferred to Simulink^®^/Simscape in a multibody environment to apply kinematic control and achieve a simulated gait. The quadrupedal form was adopted to execute tasks and avoid underwater obstacles using the dynamic advantages of this form during gait [12] (high load–weight ratio, high precision, and high resistance).

The leg mechanism was 2 D.O.F.-5R planar with four links $L_{i}$ = 150 mm (i = 1, 2), one link $L_{3}$ = 300 mm, and another link $L_{B}$ = 380 mm in the base (see Figure 1a and Figure 2). Each leg was connected to the body with a shoulder joint without movement; the main robot measurements are shown in Figure 1b. The proposed geometry offers the robot advantages for future developments for the lifting of loads, swimming, and navigation over the seabed.

### 2.2. Inverse Kinematics of 5R Leg Mechanism

The 5R mechanism had two active joints; the inverse kinematics indicated the articular positions of these from the Cartesian position P(x, y, z) [13,14] in the leg tip to the local frame origin O placed in its base (z = 0 for a planar system). In the corresponding literature, some kinematic models of planar 5R mechanisms have been reported [15,16,17]. Figure 2 shows the kinematic diagram for the computation; from the inventor geometry, the required angles and dimensions can be obtained. At point E, with trigonometric manipulations, we could obtain the following expressions: (1)$$Ex=-0.7x+0.517L_{3},$$
(2)$$Ey=-0.7y-0.85L_{3}.$$

Moreover, for the $\alpha_{i}$ and $\beta_{i}$ angles (i=1,2), we obtained: (3)$$\alpha_{i}=acos\left(\frac{{L_{1}}^{2}+{\left(L_{0}+E_{x}\right)}^{2}+{E_{y}}^{2}-{L_{2}}^{2}}{2+L_{1}\sqrt{{\left(L_{0}+E_{x}\right)}^{2}+{E_{y}}^{2}}}\right),$$
(4)$$\beta_{i}=atan\left(\frac{E_{y}}{L_{0}+E_{X}}\right),$$
where $E_{x}$ is positive for $\alpha_{1}$ and $\beta_{1}$ and negative for $\alpha_{2}$ and $\beta_{2}$.

Then, from Equations (Equation 1) and (Equation 2), we obtained the $Q_{1}$ and $Q_{2}$ articular coordinates: (5)$$Q_{1}=-\beta_{1}-\alpha_{1},$$
(6)$$Q_{2}=\beta_{2}+\alpha_{2}+180.$$

## 3. Seabed Model

The block diagram, for the seabed, was modeled with the Contact Force Library [18], with the sphere to plane force concept, including the penetration volume $V_{P}$ according to the sphere radius length (2R × 2R × *R*), knowing the value for the stiffness and damping, and for a continuous friction law, the kinetic friction and static friction coefficients. *R* was determined by the robot geometry in the leg tip (*R* = 0.031 m). In [19], the preconsolidation pressure for a terrigenous sediment seabed was $P_{C}$ = 34.47 kPa. We determined the largest overburden pressure that could be exerted on the soil without unrecoverable volume change. With these data, and the area of the robot leg tip, from its geometry $A_{tip}=7.6\times 10^{-4}$ m${}^{2}$, the limit applied force into the seabed soil could be found with Equation (Equation 7): (7)$$F_{limit}=Area_{tip}.P_{C},$$
then, this force is $F_{limit}$ = 26.2 N.

### 3.1. Dynamic Seabed Model: Mass–Spring–Damper–Friction

For the linear force law, the seabed can be modeled as a second-order system with an elastic spring and damper behavior, with friction in the leg tip traveling into the seabed, in accordance with the pseudo-rigid-body model approach [20]. Figure 3 shows the equivalent model for the mass–spring–damper–friction system (hereafter referred to as the M.S.D.F. system), and Equation (Equation 8) gives the mathematical model used in the subsequent analysis.
(8)$$m\ddot{u}+c\dot{u}+ku+\mu_{i}N=F,$$
where*m* = soil mass [Kg];*u* = displacement [m];*k* = contact stiffness coefficient [N/m];*c* = damping coefficient [Ns/m];$\mu_{i}$ = friction coefficient (*i* = s static, *i* = d dynamic, as appropriate);*F* = applied force [N];*N* = normal force [N].

The force F(t) is applied with a sinusoidal input from the robot weight divided by four (from the Inventor model) with a value $W_{Leg}$ = 47.175 N (equivalent to the bias) and added force $F_{e}$ = 1 N for eventual external disturbance (equivalent to the wave amplitude). Comparing F(t) with $F_{limit}$, we concluded that there will be deformation in the seabed soil, which can be obtained with the M.S.D.F. system modeling. The soil mass was found in [21], where the firm mud density is $\rho_{mud}=1580$ kg/m${}^{3}$. Then, the value for the penetration volume for *R* = 0.031 m is $V_{P}=0.00012$ m${}^{3}$, and the firm mud mass could be found: $m_{mud}=0.1896$ kg. For friction coefficient data, see Section 3.2. The static friction coefficient was not considered in the dynamical analysis.

Equation (Equation 8) was represented in a block diagram in Simulink to obtain its velocity response in time. Figure 4 shows the diagram model of the system with the respective variable and coefficient data. The velocity results had a sinusoidal form and were used with an integrator block to calculate the penetration depth in the seabed in the time domain, obtaining a maximum depth $P_{D}=9.78\times 10^{-8}$ m in firm mud. These results (Table 1) showed that the soil reference plane width in the simulation was >$P_{D}$.

### 3.2. Contact Force Parameters

*Contact stiffness coefficient (k):* This is the reaction force that the seabed applies per unit penetration depth (displacement *u*) in the contact area. In [22], for a chain line body, the stiffness value per unit length of the line is (k/l)=20 N/m${}^{2}$, and the corresponding coefficient for the contact depth of the sphere to plane concept for R is k=0.62 N/m.

*Contact damping coefficient (c):* This is the relation between the normal force applied when the object travels into the seabed (not when it is coming out) and the velocity normal to the seabed. In [23], using the phase-shift method in a hardened soil model with small strain stiffness, the vertical damping coefficient, C (Ns/m), was calculated from 150 to 600 kN loads at 0.6 Hz (4 rad/s; see Section 4.1: Input signals). Then, with these data and a polynomial regression method, we obtained a value of $C=1.166\times 10^{7}$ Ns/m for a leg weight $W_{leg}$ = 0.047175 N as a load.

*Kinetic and static friction coefficients $\mu_{k}$ and $\mu_{s}$:* As in the above paragraphs, the sliding friction coefficient was shown in [24]. Between a chain and seabed, the dynamic friction coefficient $\mu_{d}$ = 0.62, and the static friction coefficient $\mu_{s}$ = 1.01 for firm mud conditions.

Table 2 summarizes all contact force parameters in firm mud conditions.

The damping and contact stiffness coefficients were modeled in the respective references in a hardened soil model with slight deformation stiffness. These showed only one model for firm mud and was considered as such in this study.

Table 2’s data were input in the Simscape contact forces interface to define the seabed model with a block diagram. In Figure 5, the seabed model block shows four inputs, each corresponding to the sphere located at the leg tip to achieve sphere–plane contact. The velocity threshold in the friction tag was 0.001 m/s to ensure the lock or unlock friction contact [25], and the plane depth to the reference frame was R/2 = 0.0155 m > $P_{D}$.

## 4. Prototype Design

### 4.1. Quadrupedal Gait Simulation

The multibody model was simulated on a CPU with a sixth-generation Core i7 processor running Windows NT 10.0, and a 32 GB RAM memory board, with Simulink release 2020b and configuration parameters of a maximum step size of 0.008 and solver ode 15 s with a Simscape inventor integration. It organized the kinematic control elements throughout a block diagram (see Figure 6) with five principal parts, described below.

(1) Input signals: for x, y sinusoidal type with specific parameters. These signals were obtained after many experiments and complyiedwith two conditions: establishing an ellipsoidal trajectory in the leg tip [26], and avoiding a singular position [27] in the theoretical workspace [28]. Table 3 summarizes all signal parameters.

The experiments were performed with different values for amplitude, bias, and frequency, taking the local frame origin O placed in its base (Figure 2, z = 0 for a planar system) in each leg. The amplitude determined the stride length on the x-axis, the bias determined the leg tip height on the y-axis, and the frequency determined the linear velocity of the quadrupedal gait. These three parameters are mentioned by Catavitello et al. [29] for six dog types that were 57 cm at the withers in height and a gait speed Vc = 0.2 m/s. The experimental results reproduced these paramters. The phase parameter was out of phase in pi/2 to ensure that the gait alternation in each leg was in accordance with [29] regarding the kinematic pattern for a swim range of motion in dogs. Then, using the sensing tool in the Simscape 6-D.O.F. joint block, the linear translation velocity could be measured (V = 0.2 m/s) in the overall quadrupedal robot.

(2) MATLAB function: A code for the inverse kinematic computation that also showed their singular values in the $Q_{1}$ and $Q_{2}$ articular positions, shown as a function block.

(3) Initial positions: For previewing the simulation as a standby state, initial values in the active joints were necessary. The shoulder joints were always in zero value, and their analysis was not considered in this study.

(4) Multibody system: This was defined by exporting the inventor CAD to the Simscape interface in Simulink and verifying the corresponding active joints in each leg.

(5) Seabed model: This was used for defining the soil parameters, e.g., contact stiffness, contact damping, and static and kinetic friction coefficients (see Section 3).

With these five elements, the simulation showed the 3D gait (Figure 7). The model was simplified from the original CAD to obtain a faster response in the simulation computing time (see the video called “Robot Gait.mp4” on the Appendix A section link).

### 4.2. Gait Simulation Results

The gait pattern shown in the simulation had two principal situations: the visualization of the gait and the seabed model’s influence on it. These results were conducive to the analysis of the angular position of the active joint and the required torque to achieve the gait and exceed the requirements of damping, stiffness, and friction for the seabed model. These requirements are described below.

#### 4.2.1. Angular Position and Velocity

For the gait state, the robot has many requirements to achieve the angular position in active joints $Q_{1}$ and $Q_{2}$. For applying the input signals (x, y) on the inverse kinematic block function, the angular position was computed and is presented in Figure 8a,b, respectively. For active joints’ angular velocity, the results are shown in Figure 8c,d. Thus, from these results, we could obtain the equivalent data for the PWM signals in the motor.

Figure 8d shows the designed values for maximum amplitude in $\omega_{2}=0.4524$ rad/s (4.32 rpm).

#### 4.2.2. Torque

From the multibody simulation, the torque required in the active joint $Q_{1}$ was as shown in Figure 9a, and for active joint $Q_{2}$, as shown in Figure 9b. Torque values for the design were the maximum amplitude data in Figure 9b, with $T_{Q2}$ = 16.02 Nm.

### 4.3. Gearbox

Torque ($T_{Q2}$ = 16.02 Nm) and angular velocity ($\omega_{2}=4.32$ rpm) requirements are described in Section 4.2, and the waterproofing could achieved with a marine electric motor, selected from BlueRobotics Inc., located in Los Angeles, U.S., of the brushless type, for operation at 20VCC, and using an ESC-R1 controller [30]. Table 4 shows the principal characteristics required.

rpm = 1120 is the angular velocity of the motor, but, according to Figure 8d, for a robot gait, it was necessary to set this at 4.32 rpm in a transmission relationship i=1/6 with three similar stages. The angular output velocity from the gearbox was $\omega_{gb}=$ 5.18 rpm $\approx \omega_{2}=$ 4.32 rpm from the simulation. For the torque, with a motor power *P* = 10 W and using the angular velocity selected $\omega_{gb}$, with dimensional analysis, the final calculated torque was $T_{gb}=18.44$ Nm > T${}_{Q2}$.

For the design of a gear drive stage (with *i* = 1/6), a planetary gearbox concept was used for compact sizing. From Inventor, the Spur Gear toolbox was used to design the gearbox stage, with a central sun gear wheel, three planetary gears, and a crown (ring gear). The parameters for the design are shown in Table 5, and the dimensions are shown in Figure 10. Each gearbox stage was printed with ASA material, assembled with stainless steel bolts and nuts, and driven by the M200 brushless motor.

### 4.4. Leg Prototype

The leg prototype was created with ASA-printed parts for the links of the 5R mechanism and the base link with a high-density polycarbonate sheet with holes for the gearbox shafts.

The leg assembly was created on two linear carriages (lineal and vertical) and on an aluminium profile frame with their respective linear guides [31]; then, they were coupled on a steel frame. Figure 11a shows the mounted prototype for the testbed (see the video called “Leg system move.mp4” on the Appendix A section link).

### 4.5. Hardware and Software Architecture

The angular position of motor θ was sensed by an encoder module consisting of a magnetic actuator and a separate sensor board, with a chip mounted on the motor shaft to achieve low-level control. The selected encoder was RMB20IC—Incremental, from RLS^®^ [32], with a square wave differential line driver to RS422; see Figure 11b. The motor had a basic speed controller ESC-R1 unit, connected to an Arduino Mega 2560 board output for processing the encoder signal and controlling its angular position. Figure 12 represents the hardware and software architecture for the leg position control. The application worked with IDE Arduino V1.8.57.0, and state estimation was managed from a laptop CPU with a sixth-generation Core i7 processor running Windows^®^10 Pro, which sent a signal from the serial COM port to the Arduino board at 115,200 Bps.

### 4.6. Low-Level Control of the Motor

From reference [30], the equivalent PWM data were used to calculate a ninth-order equivalent polynomial regression (with MSE = 4.91 error) for angular velocity $\dot{\theta }$ (in revolutions per minute) converted to PWM data with the form: (9)$$P\left(x\right)=p_{1}x^{9}+p_{2}x^{8}+p_{3}x^{7}+\ldots +p_{9}x+p_{10},$$
where $p_{1}=-4.873\times 10^{-31}$,  $p_{2}=-5.102\times 10^{-28}$,  $p_{3}=4.09\times 10^{-23}$,

  $p_{4}=1.817\times 10^{-20}$,  $p_{5}=-9.129\times 10^{-16}$,  $p_{6}=-3.2\times 10^{-13}$,

  $p_{7}=8.641\times 10^{-09}$,  $p_{8}=1.638\times 10^{-06}$,  $p_{9}=0.0662$,  $p_{10}=1501$.

The generated curve is shown in Figure 13.

Inverse kinematic block results sent the angular position signal multiplied by a constant $6^{3}$ (by three gearbox stages). This signal was used to obtain the angular velocity (see Figure 14). With Equation (Equation 9), the PWM data were obtained for the setpoint.

The final PWM curves obtained had an approximate sine waveform with the parameters in Table 6.

Notably, the input PWM parameter to the controller ESC was a result of a conversion in the time domain. Using it as an input to the motor also resulted in a “$PWM_{controller}$ value” as a function of time.

The implemented algorithm for the angular position control used the PID concept. It sought to keep the input variable close to the desired setpoint signal by adjusting the output by tuning the PID parameters. Then, using the Simulink auto-tuning tool, we obtained the closer desired output and implemented it on the Arduino program for $K_{P}$ = 0.24, $K_{I}$ = 0.265 and $K_{D}$ = 0.0029 for $Q_{1}$, and $K_{P}$ = 0.18, $K_{I}$ = 0.27 and $K_{D}$ = 0.0042 for $Q_{2}$.

The motor–ESC group worked as a black box (plant model) and complied with the classical closed-loop controller. Thus, by introducing a PWM input signal to the motor, an approximately angular velocity response was obtained at the active joint that the encoder sensed to feedback the loop [33]. Path planning control at the leg tip was not considered in this study. Figure 15 shows the closed-loop control system.

## 5. Results

Angular velocity control followed by angular position control in active joints were tested in an underwater environment (see Figure 16 and the video called “Testbed.mp4” on the Appendix A section link).

The setpoint signal and angular position θ response in the time domain for $Q_{1}$ and $Q_{2}$ are shown in Figure 17 and Figure 18, with a mean squared error $MSE_{Q_{1}}=1.743\times 10^{-7}$ and $MSE_{Q_{2}}=5.4184\times 10^{-7}$, respectively.

For angular velocity $\dot{\theta }$, the simulation results and the Arduino program response in the time domain for $Q_{1}$ and $Q_{2}$ are shown in Figure 19 and Figure 20, respectively.

## 6. Discussion

### 6.1. Gait Simulation

The simulation model provided an adequate approximation for determining the influence of many variables, positions, and geometry in the quadrupedal gait and then for calculating torque. The multibody model needed an adequate computer system with many hardware resources to solve its dynamics. Referring to the specialized literature helped us to identify many characteristics of the real environment and place them in the virtual model. The multibody systems could be simulated with other input signals to obtain other gait and movement states. Different authors have reported different parameters for damping and contact stiffness coefficients that can determine a simulation with different position and torque results. An alternative would be to perform the experimental calculation of these coefficients using soil models with materials similar to those determined in the literature. Regarding the quadruped type of gait, other states of motion can be obtained and used to achieve other goals in path planning and high-level control considering the dynamic effects.

### 6.2. Angular Position Control of Motors

From the simulation results, the normal gait state presented different modes for movement from the eight active joints (the shoulder joint was not analyzed in this study), but it could be normalized by sending the correct input signals to the motors. The M200 motor was probed as a thruster, in use on thousands of marine robotic vehicles around the world, with a speed controller, but not for angular position control. The angular position control results from the velocity control showed a different application in these novel robot systems. Moreover, the testbed angular position results showed unusual behavior in the maximum wave peak for $Q_{1}$ and at the minimum peak for $Q_{2}$. From the MSE information, higher error occurred in the $Q_{2}$ position, but distortion could be caused by the backlash in the angular position of the motor gearbox system at the time of landing the leg tip on the soil; this hypothesis should be verified by future experimentation.

### 6.3. Velocity Control of Motors

The designed PID controller for the motor achieved an adequate amplitude and frequency response and could be tuned with other methods. The backlash effect could be considered a mechanical impedance and could produce angular position and velocity errors, as can be seen in Figure 19 and Figure 20, through the PWM velocity response. In addition, the disturbances caused in the angular position signals may have been due to harmonics produced by the variable-speed electronic drive. The possible implementation of filters to eliminate parasitic signals should be analyzed.

## 7. Conclusions and Future Works

The design of an alternative system for quadrupedal gait was developed and tested in a specific underwater environment in which the hydrodynamic damping effects did not need to be considered.A single method was constructed that was contrasted with a computational model.

Initially, we built a simulation model with the most realistic parameters for the soil and quadrupedal gait throughout a multibody model that provided initial information about the gait pattern produced to introduce an adequate input signal.

The sine wave signal required (Table 3) for reproducing the normal gait was found after many experiments to avoid singular positions of the 5R parallel mechanism. We observed the relationship between the ellipsoidal trajectory characteristics described by the leg tip and the gait type, as well as the linear velocity and torque required in each experiment to reproduce a pattern that approximated a dog’s gait.

The seabed model was simulated by introducing it into a multibody system with the mechanical properties and characteristics of the soil in analysis via information obtained from the related literature. The deep penetration length was obtained by a second-order mass–spring–damper–friction system (M.S.D.F.) and yielded a maximum depth $P_{D}=9.78\times 10^{-8}$ m in firm mud.

From the gait simulation results, the angular velocity required was $\omega_{2}=0.4524$ rad/s (4.3201 rpm), as shown in Figure 8d, and the torque results in Figure 9 presented higher peaks in $Q_{2}$ displacement with $T_{Q2}=16.02$ Nm.

A motor–gearbox group (see Section 4.3) was designed, made, and probed for the specific requirements of angular velocity and torque in the underwater environment and could be improved for decreasing the backslash between gears. Furthermore, it can be considered a device with patent potential for new developments in the underwater exploration given its characteristics of small sizing, high torque, and low angular velocity.

The hardware and software architecture (Section 4.5) were designed according to low-weight and high-performance requirements to implement low-level control in each motor. The angular position control was achieved through velocity control with the manufacturer’s data for the equivalent PWM to velocity in revolutions per minute and introducing a PID controller in a closed-loop configuration, as well as the use of an encoder as a sensor to transform the velocity in the angular position, which we compared with a sinusoidal setpoint provided by the simulation.

The angular position testbed results, shown in Figure 17 and Figure 18, indicated an unusual perturbation in the contact point between the leg tip and soil, in the maximum wave peak for $Q_{1}$, with a mean squared error $MSE_{Q_{1}}=1.743\times 10^{-7}$, and a minimum peak for $Q_{2}$ and $MSE_{Q_{2}}=5.4184\times 10^{-7}$.

Velocity control resulted in an adequate amplitude and frequency response and could be tuned with other methods and probed with other control models. A complete underwater quadrupedal robot can be developed based on these initial results for many tasks in this environment.

The influence of the shoulder joint should also be analyzed in future studies to improve the gait pattern and extend the research toward robot trajectory control (high-level control).

The 5R geometry provides an alternative for mobility in underwater environments if a membrane is added between its links for swim tasks.

## Figures and Tables

**Figure 1 sensors-22-08462-f001:**
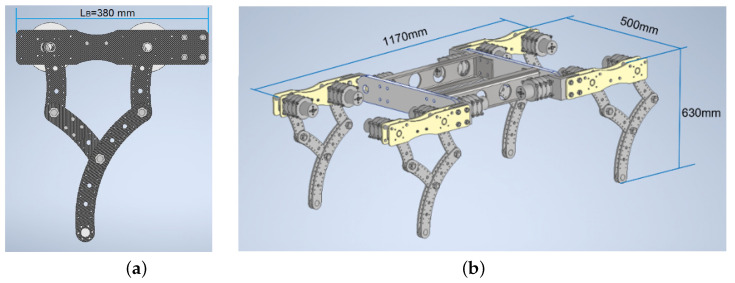
(**a**) Leg mechanism 2 D.O.F.-5R. (**b**) Robot testbed.

**Figure 2 sensors-22-08462-f002:**
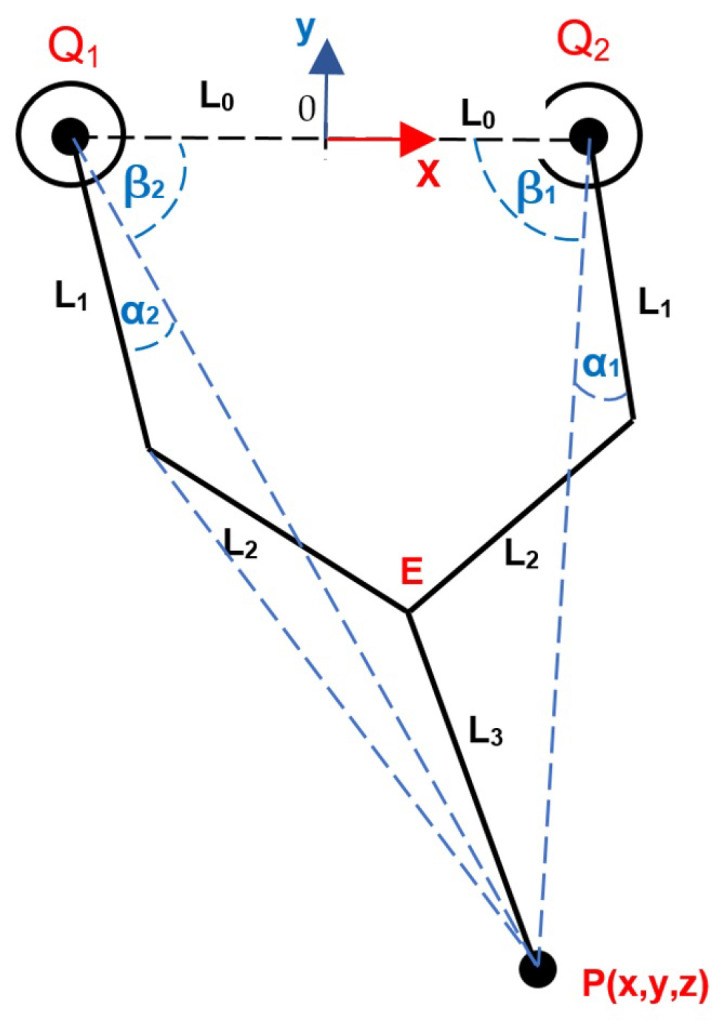
Kinematic diagram of the leg.

**Figure 3 sensors-22-08462-f003:**
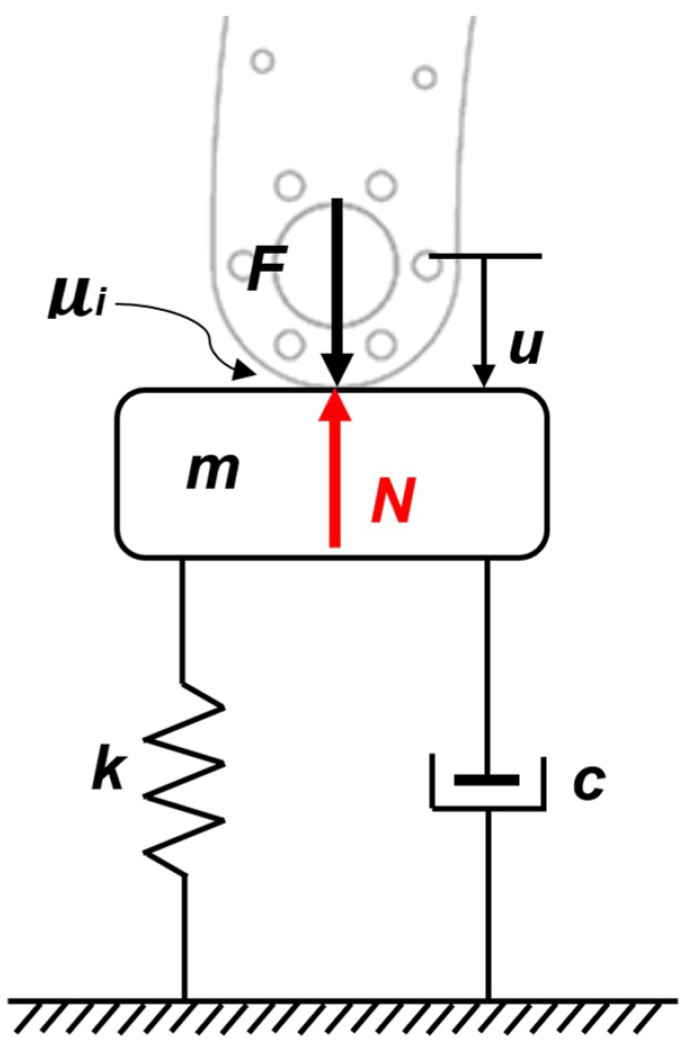
Equivalent mass–spring–damper–friction (M.S.D.F.) system for the seabed.

**Figure 4 sensors-22-08462-f004:**
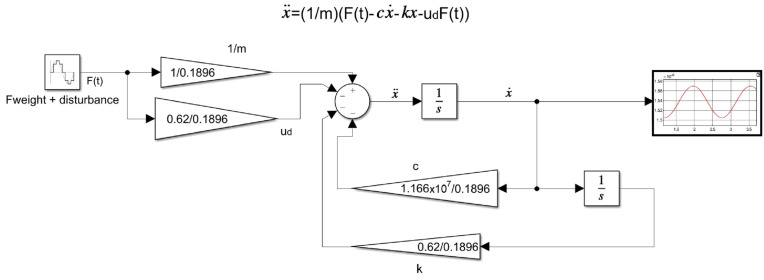
Block diagram for the M.S.D.F. equivalent system in firm mud soil.

**Figure 5 sensors-22-08462-f005:**
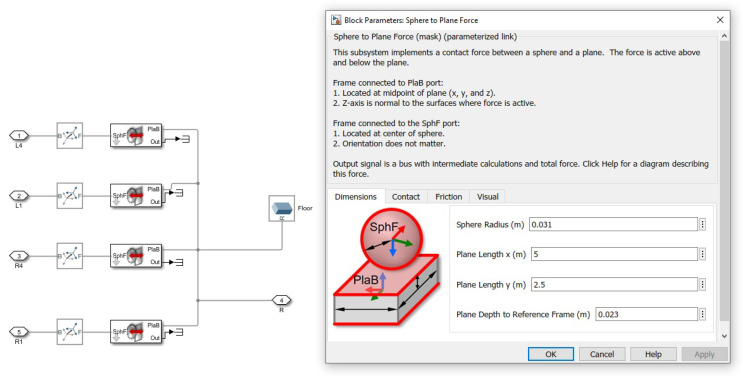
Simscape block parameters for sphere-to-plane contact.

**Figure 6 sensors-22-08462-f006:**
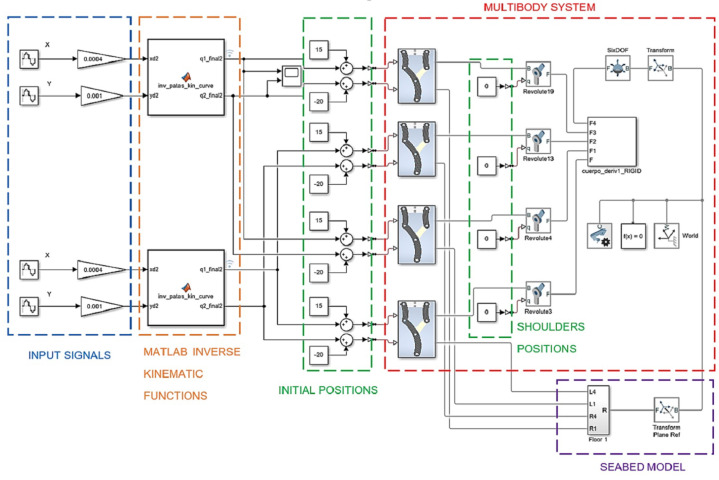
Block diagram for quadrupedal gait.

**Figure 7 sensors-22-08462-f007:**
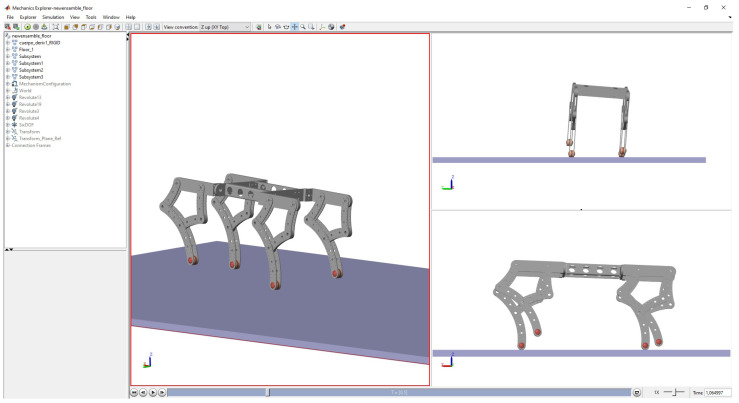
Quadrupedal gait 3D interface.

**Figure 8 sensors-22-08462-f008:**
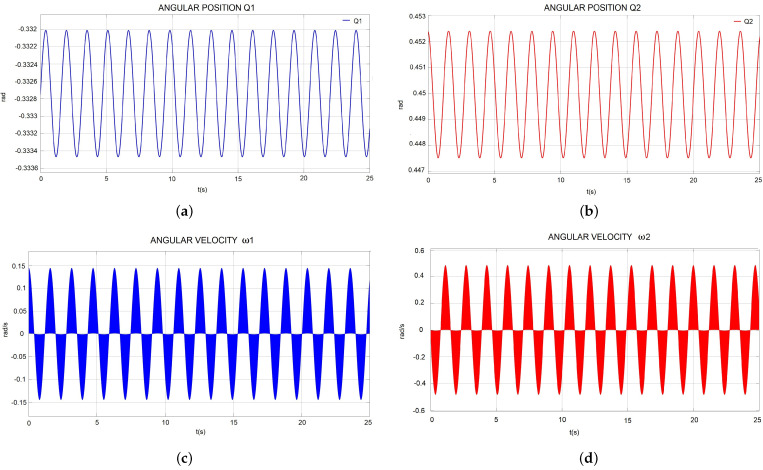
(**a**) Angular position $Q_{1}$. (**b**) Angular position $Q_{2}$. (**c**) Angular velocity $\omega_{1}$. (**d**) Angular velocity $\omega_{2}$.

**Figure 9 sensors-22-08462-f009:**
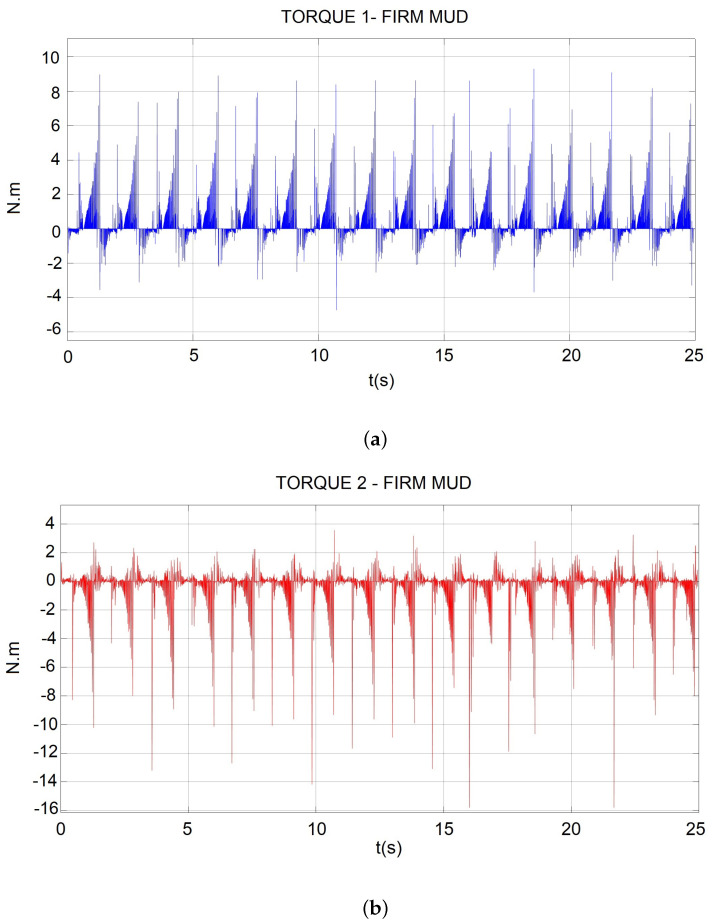
(**a**) Required torque in the active joint $Q_{1}$; (**b**) required torque in the active joint $Q_{2}$.

**Figure 10 sensors-22-08462-f010:**
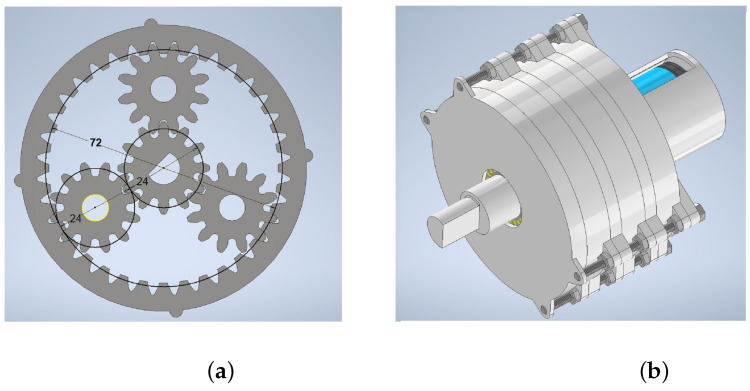
(**a**) Dimensions of the planetary gear set. (**b**) Three stages of planetary gearbox assembly.

**Figure 11 sensors-22-08462-f011:**
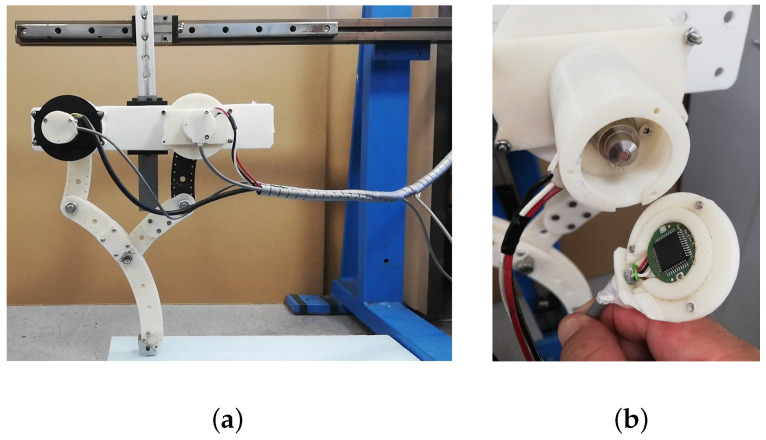
(**a**) Assembled leg prototype. (**b**) Encoder mounted on the motor.

**Figure 12 sensors-22-08462-f012:**
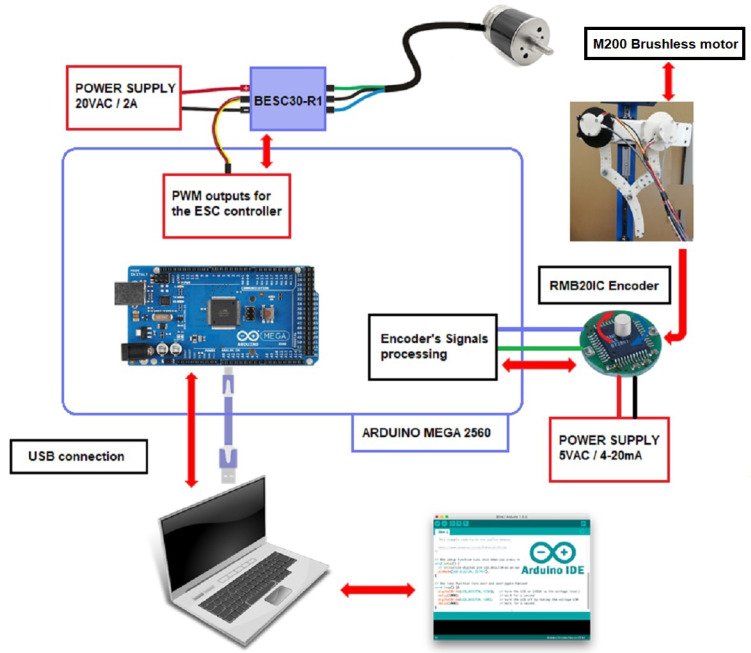
Hardware and software architecture for angular position control.

**Figure 13 sensors-22-08462-f013:**
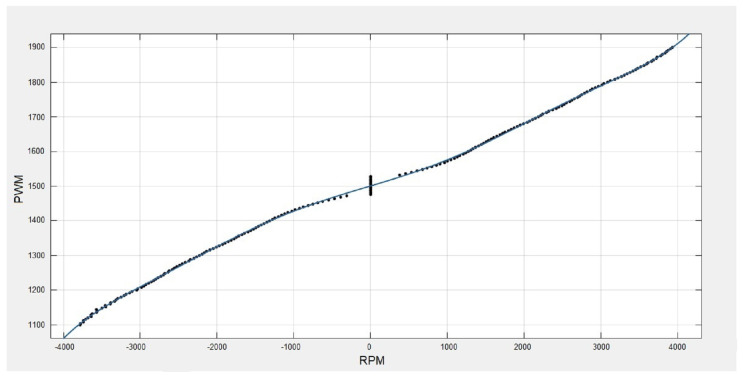
Polynomial regression curve: rpm–PWM.

**Figure 14 sensors-22-08462-f014:**
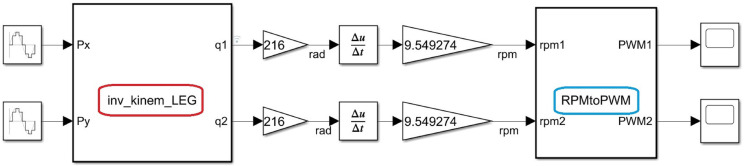
rpm to PWM conversion for angular position.

**Figure 15 sensors-22-08462-f015:**
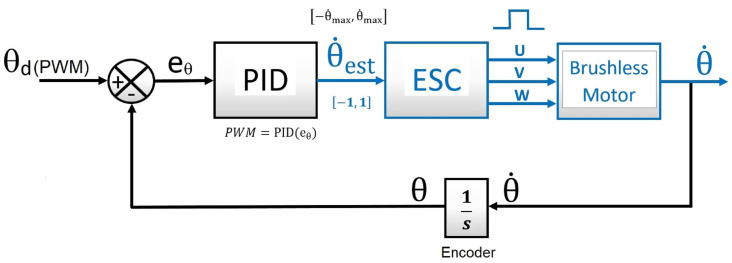
PID closed loop for angular position control in an active joint.

**Figure 16 sensors-22-08462-f016:**
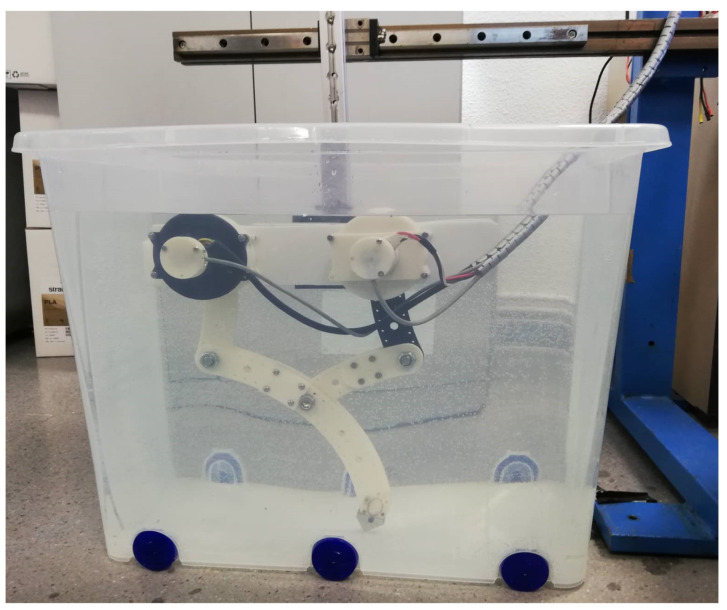
Prototype testbed in underwater environment.

**Figure 17 sensors-22-08462-f017:**
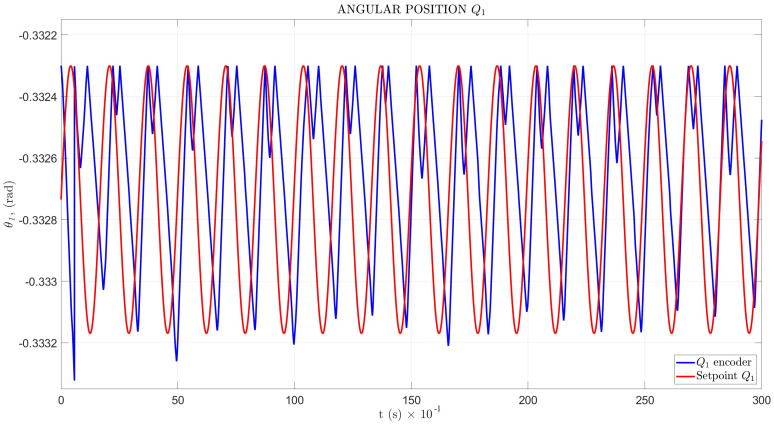
Angular position for $Q_{1}$.

**Figure 18 sensors-22-08462-f018:**
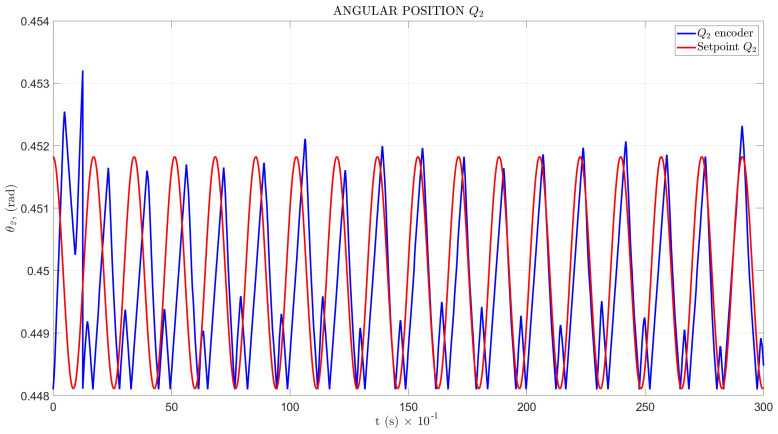
Angular position for $Q_{2}$.

**Figure 19 sensors-22-08462-f019:**
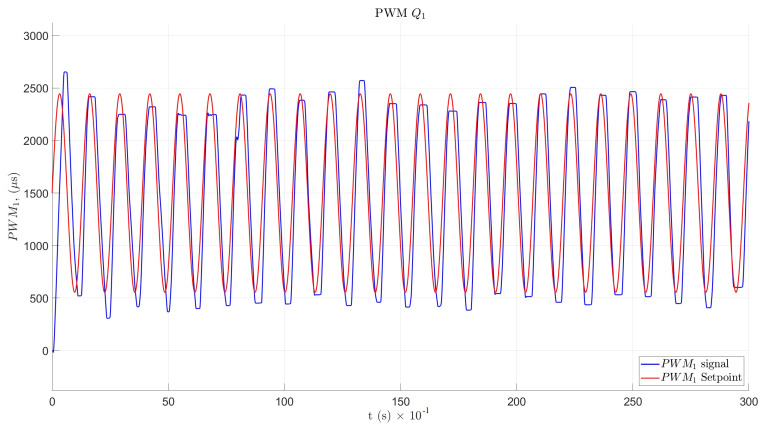
PWM velocity response for $Q_{1}$.

**Figure 20 sensors-22-08462-f020:**
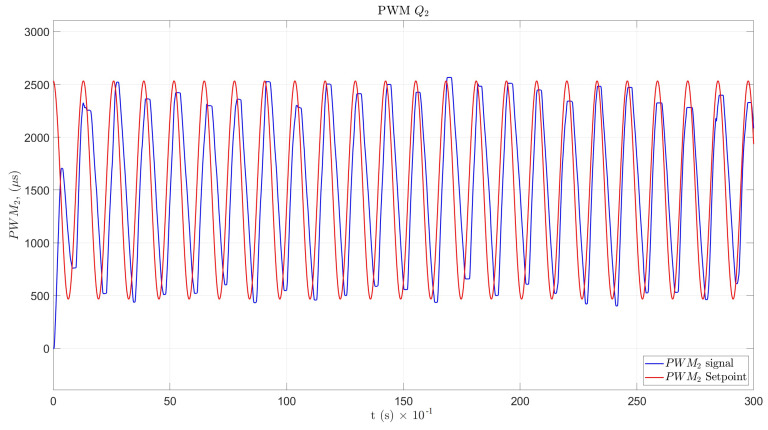
PWM velocity response for $Q_{2}$.

**Table 1 sensors-22-08462-t001:** Velocity and depth results.

Soil	Vmax–Vmin (m/s)	tmax–tmin (s)	Depth (m)
Firm mud	$6.52\times 10^{-8}$	1.5	$9.78\times 10^{-8}$

**Table 2 sensors-22-08462-t002:** Contact force parameters for two seabed conditions.

Coefficient	Firm Mud
Contact stiffness *k* (N/m)	0.62
Contact damping *c* (Ns/m)	$1.166\times 10^{7}$
Dynamic friction ($\mu_{d}$)	0.62
Static friction ($\mu_{s}$)	1.01

**Table 3 sensors-22-08462-t003:** Input signal parameters.

Parameters	X Coordinate	Y Coordinate
Amplitude (m)	$-4.2\times 10^{-4}$	$0.5\times 10^{-3}$ m
Bias (m)	$-6.8\times 10^{-4}$ m	$-1.2\times 10^{-3}$ m
Frequency (rad/s)	4	4
Phase (rad)	pi/2	0
Sample time (s)	0.008	0.008

**Table 4 sensors-22-08462-t004:** Performance data for BlueRobotics motor [30].

PWM (μs)	rpm	Current (A)	Voltage (V)	Power (W)	Force (kgf)	Effic. (g/W)
1420	1120	0.5	20	10	−0.39	38.6
1500	0	0	20	0.0	0.00	0.0
1580	1090	0.5	20	10.0	0.47	46.7

**Table 5 sensors-22-08462-t005:** Parameters for gearbox stage design.

Parameter	Crown	Sprocket
Pitch diameter Pd (mm)	72	24
Whole depth h (mm)	7.58	9.2
No. teeth Z	36	12
Thickness e (mm)	5	5

**Table 6 sensors-22-08462-t006:** PWM sine wave signal parameters.

Parameter	Amplitude (μs)	Bias (μs)	Frequency (rad/s)	Phase (rad)
PWM $Q_{1}$	4729.752	1500	0.374	0
PWM $Q_{2}$	1034.004	1500	0.374	pi/2

## Data Availability

Not applicable.

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
