# Peer review of "Method to Develop Legs for Underwater Robots: From Multibody Dynamics with Experimental Data to Mechatronic Implementation"

_sensors, 2022, doi:10.3390/s22218462_

Round 1
Reviewer 1 Report
Title: Method to develop legs for underwater robots: from multibody dynamics with experimental data to mechatronic implementation
Authors: MIGUEL ÁNGEL PÉREZ BAYAS, JUAN CELY, AVISHAI SINTOV, CECILIA GARCÍA and ROQUE SALTAREN
ID: sensors-1931806
The paper presents the gait simulation of a quadrupedal robot used on a typical terrigenous sediment seabed, taking into consideration mechanical properties, stiffness and damping and friction coefficients, in a specific underwater environment. The paper represents research related to a practical application. The scientific novelty is reduced, but the interest in the domain has motivated to discover and explore more areas of research by the authors' determination to make a complete study of the problem can be appreciated. However, the paper is correctly structured and well written and presents the authors' ability to operate with numerical methods in solving such problems. As results I can recommend the acceptance of this paper after some corrections.
The abstract needs to be improved because it provides too much information which is then repeated in the paper. So, the abstract must contain, in summary, what was done in the paper, without discussions or details from the work.
Authors should improve the Introduction parts and the references needs to present the background of the research. At the end of this section must be established more clear the original contribution of the authors.
Punctuations are used randomly. Insert comma, semicolon or full stop every time when is necessary. After each mathematical relation a such sign must exist.
The section Conclusions will be point out the original results of the paper and can be extended to highlight the contributions.
The paper must be improved in order to explain more clear the contribution of the author and the existing state of art.
I think the authors need to emphasize more clearly the contribution of the manuscript from a scientific point of view.
If the author take these into consideration, I recommend the acceptance of this paper.
Reviewer 2 Report
This paper introduces a method to develop a leg system for an underwater robot. The paper is well-written and has clearly structured its contents. This article introduces the model of the leg system and its dynamics, and the analysis of the gait simulation on the seabed is also given. The leg prototype is also given with its hardware and software architecture for the low-level control of motors. The results can confirm the leg system can work in the underwater environment.
Major:
======
1. Method and Novelty:
----------------------
The contribution of this work should be listed more clearly. *when describing the design*, e.g. what is the difference between this design when compared with the previous work? What is the main challenge to design a leg for the underwater robot? I highly suggest authors list the contributions more clearly in the section of the introduction.
2. Evaluation:
--------------
I appreciate authors provide the leg prototype with its motor system, however, it would be better to show a video of the movement of the leg system. A deeper discussion of the simulation results should also be given, e.g. in Figures 17, 18, and 20, there are irregular movements at the beginning of the time. What is the reason?
Minor:
======
1. In line 77, where is L_B? it is better to add the length of these links in Figure 1a or Figure 2.
2. It would be better for authors to avoid short paragraph, e.g. line 80-81, and line 88-89.
3. In line 85, it is better to use a character instead of a number to describe the frame origin.
